miR-1293 acts as a tumor promotor in lung adenocarcinoma via targeting phosphoglucomutase 5

Chen Bing 1 2
Zheng Shiya 3
Jiang Feng 1 2 fengjiang_nj@njmu.edu.cn
1 Department of Thoracic Surgery, The Affiliated Cancer Hospital of Nanjing Medical University & Jiangsu Cancer Hospital , Nanjing , China
2 Jiangsu Key Laboratory of Molecular and Translational Cancer Research, Cancer Institute of Jiangsu Province , Nanjing , China
3 Department of Oncology, Zhongda Hospital, School of Medicine, Southeast University , Nanjing , China
Uversky Vladimir
Electronic publication date: 2021 Sep 16
Publication date: 2021
Volume: 9
Electronic Location ID: e12140
Received 2021 May 27; Accepted 2021 Aug 19
Copyright: © 2021 Chen et al.
Copyright year: 2021
Copyright holder: Chen et al.
License: This is an open access article distributed under the terms of the Creative Commons Attribution License, which permits unrestricted use, distribution, reproduction and adaptation in any medium and for any purpose provided that it is properly attributed. For attribution, the original author(s), title, publication source (PeerJ) and either DOI or URL of the article must be cited.
License URL: https://creativecommons.org/licenses/by/4.0/

Keywords: miR-1293, Lung adenocarcinoma, Phosphoglucomutase 5, Migration, Invasion

Funding: Youth Foundation of Jiangsu Province BK20200395 The study was supported by the Youth Foundation of Jiangsu Province (BK20200395). The funders had no role in study design, data collection and analysis, decision to publish, or preparation of the manuscript.

==============================
Background

Lung adenocarcinoma (LUAD) is the most common histologic subtype of lung cancer. Studies have found that miR-1293 is related to the survival of LUAD patients. Unfortunately, its role in LUAD remains not fully clarified.

Methods

miR-1293 expression and its association with LUAD patients’ clinical characteristics were analyzed in TCGA database. Also, miR-1293 expression was detected in LUAD cell lines. Cell viability, migration, invasion and expression of MMP2 and MMP9 were measured in LUAD cells following transfection with miR-1293 mimic or antagomir. Phosphoglucomutase (PGM) 5 was identified to be negatively related to miR-1293 in LUAD patients in TCGA database, and their association was predicated by Targetscan software. Hence, we further verified the relationship between miR-1293 and PGM5. Additionally, the effect and mechanism of miR-1293 were validated in a xenograft mouse model.

Results

We found miR-1293 expression was elevated, but PGM5 was decreased, in LUAD patients and cell lines. Higher miR-1293 expression was positively related to LUAD patients’ pathologic stage and poor overall survival. miR-1293 mimic significantly promoted, whereas miR-1293 antagomir suppressed the viability, migration, invasion, and expression of MMP2 and MMP9 in LUAD cells. PGM5 was a target of miR-1293. Overexpression of PGM5 abrogated the effects of miR-1293 on the malignant phenotypes of LUAD cells. Administration of miR-1293 antagomir reduced tumor volume and staining of Ki-67 and MMP9, but elevated PGM5 expression in vivo.

Conclusions

miR-1293 promoted the proliferation, migration and invasion of LUAD cells via targeting PGM5, which indicated that miR-1293 might serve as a potential therapeutic target for LUAD patients.

Introduction

Lung cancer is the leading cause of cancer-related death worldwide (Bray et al., 2018). Lung cancer is divided into two major subtypes, non-small cell lung cancer (NSCLC) and small cell lung cancer (SCLC), which account for 85% and 15% of all cases, respectively. NSCLC is composed of three types: adenocarcinoma (40%), squamous cell carcinoma (25–30%), and large cell carcinoma (10–15%) (Read et al., 2004). This indicates that lung adenocarcinoma (LUAD) is the most common histologic subtype of lung cancer. LUAD originates from small airway epithelial and type II alveolar cells, which secrete mucus and other substances (Denisenko, Budkevich & Zhivotovsky, 2018). The 5-year overall survival rate of patients with LUAD is relatively low, with only about 18.1%. This may be attributed, its part, to the complexity of its molecular etiology (Alam et al., 2018; Cancer Genome Atlas Research Network, 2014). Hence, it is of great necessity to have a better understanding of the mechanisms of LUAD tumorigenesis.

MicroRNAs (miRNAs) are a class of small (18–22 nucleotides), single-stranded, endogenous non-coding RNA molecules. miRNAs are often aberrantly expressed and act as tumor promotors or suppressors in many different types of cancers including lung cancer (Davalos et al., 2020; Kandettu et al., 2020; Wu et al., 2019). Studies have verified that miRNAs negatively mediate gene expression at the posttranscriptional level via binding to the 3′-untranslated regions (UTRs) of target mRNAs through specific base pairing (Du et al., 2020; Kandettu et al., 2020). A recent study has found that miR-1293 expression is significantly related to the survival of LUAD patients (Wang et al., 2020). However, its effects on the malignant phenotypes of lung cancer especially LUAD are unclear.

In the present study, we analyzed miR-1293 expression in patients with LUAD and lung squamous cell carcinoma (LUSC) using data from The Cancer Genome Atlas (TCGA) database. The influences of miR-1293 on cell proliferation, migration and invasion were investigated using two LUAD cell lines by the treatment of miR-1293 mimic or antagomir. Importantly, the specific mechanism involving in tumor-promoting role of miR-1293 was further explored. Additionally, the role and mechanism of miR-1293 in LUAD were further confirmed in a mouse xenograft model.

Materials & Methods

Analysis of TCGA database and bioinformatics analysis

The expression levels of miR-1293 and phosphoglucomutase (PGM) 5 in LUAD and/or LUSC patients were analyzed in TCGA database using Starbase v3.0 (https://bio.tools/starbase) and/or GEPIA2 (http://gepia2.cancer-pku.cn/) online software. The correlation between miR-1293/PGM5 and patients’ pathologic stage, overall survival, or frequent mutations (eg. KRAS, EGFR, ALK and PIK3CA) as well as the relationship between miR-1293 and PGM5 were assayed in TCGA database via linkedomics (http://www.linkedomics.org/), Starbase v3.0 or GEPIA2 software. The correlations between miR-1293 and these mutations were analyzed by Wilcoxon test. Survival analysis curves were obtained using a Log rank Mantel cox test. RNA expression profiles in GSE118370 were downloaded from the Gene Expression Omnibus (GEO) database (https://www.ncbi.nlm.nih.gov/geo/), and differentially expressed genes in LUAD tissues were screened by GEO2R with thresholds of |log2 FC (fold-change)| > 2 and P < 0.01. The target genes of miR-1293 were analyzed by TargetScan (version 7.2).

Cell culture

This study used three LUAD cell lines A549, H1299 and H1975, two LUSC cell lines H226 and H520, and one normal lung cell line BEAS-2B. H520 cells were from the American Type Culture Collection (ATCC; Manassas, VA, USA), and other cell lines were obtained from the Cell Bank of the Chinese Academy of Sciences (Shanghai, China). They were all grown in RPMI-1640 medium, containing 10% fetal bovine serum (FBS; Gibco, MD, USA) and 1% penicillin-streptomycin (Gibco) at 37 °C with 5% CO2.

Plasmid construction and cell transfection

For the PGM5 overexpression study, PGM5 cDNA amplified from BEAS-2B cells was cloned into the pcDNA3.1 vector to generate pcDNA3.1-PGM5 plasmid. All the transfection processes were carried out with Lipofectamine 3000 (Invitrogen, Carlsbad, CA, USA). A total of 2 µg of constructed vector or pcDNA3.1 empty plasmid was transfected into H1299 cells. miR-1293 mimic, miR-1293 antagomir (anti-miR-1293), and their negative controls were synthesized by RiboBio co., LTD (Guangzhou, China). A total of 50 nM of miR-1293 mimic or 100 nM of miR-1293 antagomir was transfected into H1975 and H1299 cells.

Dual-luciferase assay

The 3′-UTR sequences of PGM5 containing miR-1293 binding sites were amplified from BEAS-2B cells and inserted into the pGL3 reporter vector (Promega, Madison, WI, USA). Also, the mutant seed region of PGM5 was cloned into the pGL3 vector. H1975 cells were plated in a 24-well plate and then transfected with miR-1293 mimic or control at a final concentration of 50 nM, and 25 ng of generated PGM5 wild-type (PGM5-WT) or mutant luciferase reporter (PGM5-MUT) using Lipofectamine 3000. A total of 48 h after transfection, cells were harvested to detect luciferase activity using a Dual-Luciferase Reporter Assay System (Promega, Madison, WI, USA).

Cell proliferation analysis

Cell Counting Kit-8 (CCK-8) was used to examine the proliferation of H1975 and H1299 cells. Cells transfected with indicated vectors were seeded into 96-well plates at a density of 1 × 103 cells/well. After culturing for 0, 24, 48 and 72 h, respectively, 10 µL of the CCK-8 solution was added to each well. A total of 2 h later, the optical density (OD) value was read by a micro-plate reader (Bio-Tek, Winooski, VT, USA) at 450 nm.

Transwell migration and invasion assay

The migration and invasion of H1975 and H1299 cells were determined by transwell analysis (Yu et al., 2020). For migration assay, the transfected H1975 and H1299 cells were trypsinized and resuspended in RPMI 1640 medium without FBS to adjust the density to 1 × 105 cells/mL, respectively. In the following, 200 μL of the cell suspension without FBS was placed into the upper chamber of transwell (8 μm in pore size, Millipore, Burlington, MA, USA), whereas the lower chamber was filled with 600 μL of RPMI 1640 medium supplemented with 10% FBS. After incubating at 37 °C for 48 h, the upper surface of the membrane was cleaned by cotton swabs, and cells on the lower surface were fixed and stained with 0.1% crystal violet for 30 min. After drying, stained cells were photographed by a light microscope. For invasion analysis, the upper members of transwell inserts were pre-coated with Matrigel, and other operation processes were consistent with the transwell migration assay.

Animal experiments

Twelve 6–8-week-old BALB/c nude mice were obtained from the Shanghai SLAC Laboratory Animal Co., Ltd (Shanghai, China). All mice were raised in specific-pathogen-free conditions under 12/12 cycle of light at room temperature (25–27 °C) and allowed free access to food and water. To investigate the role of miR-1293 in the tumorigenesis of LUAD, H1975 cells (2 × 106) were subcutaneously injected into the right flank of each mouse. One week later, mice were randomly divided into two groups (n = 6 per group): anti-miR-1293 group and anti-NC group. Mice were intratumorally injected with 5 nM miR-1293 antagomir or control, respectively, twice a week for 3 weeks. Tumor volumes were measured every week and calculated with the formula V (mm3) = 1/2 (length × width2). A total of 5 weeks after inoculation, mice were sacrificed by cervical dislocation after inhalational anesthesia with 2% isoflurane, and the xenografted tumors were removed for analysis. A part of tumor tissues was directly stored at −80 °C, which was used to detect the expression of miR-1293 and PGM5. The other part of tumor specimens was fixed in formalin and embedded in paraffin to perform immunohistochemical staining for Ki-67 and MMP9. All procedures involving in animals were reviewed and approved by the Affiliated Cancer Hospital of Nanjing Medical University (No. 2020-005).

Real-time PCR

Total RNA was isolated from cells and tissues using TRIzol reagent (Invitrogen, Carlsbad, CA, USA) following the manufacturer’s instructions. cDNA was synthesized by reverse transcription with the reverse transcription kit (Takara, Dalian, China). The relative levels of PGM5 transcripts were detected by Real-time PCR using the SYBR Green mix (Kakara, Dalian, China). The expression of miR-1293 was determined by TaqMan MicroRNA assay kits (Applied Biosystems, Foster City, CA, USA). The relative fold changes of candidate genes were assayed by the 2−ΔΔCt method, with GAPDH or U6 as an internal control. The primers used were as follows: PGM5, forward, 5′- GATGCTGATGGGGACCGTTA-3′ and reverse, 5′-GCAACGTCCTGAGTCCATCA-3′; GAPDH, forward, 5′-GAAGACGGGCGGAGAGAAAC-3′ and reverse, 5′-CCCAATACGACCAAATCCGTTG-3′; miR-1293, forward, 5′-ACACTCCAGCTGGGTGGGTGGTCTGGAGAT-3′ and reverse, 5′-TGGTGTCGTGGAGTCG-3′; U6, forward, 5′-CTCGCTTCGGCAGCACA-3′ and reverse, 5′-AACGCTTCACGAATTTGCGT-3′.

Western blot

Cells with different treatments and tumor tissues were lysed with RIPA lysis buffer (Sigma–Aldrich, St. Louis, MO, USA) supplemented with protease inhibitors. Protein concentration was quantified using the BCA Protein Assay Kit (Pierce, Rockford, IL, USA). Samples were denatured by heating in a water bath at 100 °C for 5 min. Cellular proteins were separated using SDS-PAGE by 10% running gel. After transferring proteins onto the PVDF membrane, the membrane was blocked with 5% defatted milk for 1 h at room temperature, incubated with primary antibodies, including PGM5 (1:500; Novus, St. Louis, MO, USA), MMP2 (1:1,000; Cell Signaling Technology, Danvers, MA, USA) and MMP9 (1:1,000, Cell Signaling Technology, Danvers, MA, USA) overnight at 4 °C, and probed with horseradish peroxidase (HRP)-conjugated secondary antibody (1:5,000). After that, the blots were detected using enhanced chemiluminescent reagents. The relative levels of proteins were analyzed by densitometric analysis using ImageJ software, with GAPDH as an internal control.

Statistical analysis

All experiments were performed in triplicates and mean ± standard deviation (SD) was calculated. Data were analyzed by the Graphpad Prism 5 software. Student’s t test and one-way analysis of variance (ANOVA) following Bonferroni or Dunnett’s post-test analysis were used to perform all statistical analyses. A P value of <0.05 is considered statistically significant.

Results

miR-1293 expression is elevated in tumor tissues of lung adenocarcinoma patients

miR-1293 expression was assayed in TCGA database using Starbase online software, and the results indicated that compared with normal samples, miR-1293 expression was upregulated in NSCLC samples, including adenocarcinoma (P = 0.0065) and squamous cell carcinoma (P = 2.0 × 10−12) (Figs. 1A and 1B). Furthermore, miR-1293 expression level was related to the pathologic stage of LUAD patients (Fig. 1C: N = 445; P = 1.916 × 10−5). Higher miR-1293 expression was correlated with poor overall survival of LUAD patients (Fig. 1D: N = 430; P = 2.621 × 10−6 or Fig. 1E: N = 503; P = 2.5 × 10−5). However, we did not find any correlation between miR-1293 expression and the pathologic stage (Fig. 1F: N = 339; P = 0.7166) or overall survival (Fig. 1G: N = 325; P = 0.7199) of LUSC patients. Specially, we further investigated the correlation between miR-1293 expression and the frequent mutations present in LUAD patients and found that miR-1293 expression was positively related to KRAS and PIK3CA mutations, but negatively related to EGFR mutation in LUAD patients, without any correlation with ALK mutation (Figs. 1H–1K). miR-1293 expression was notably elevated in LUAD cell lines, including A549, H1299 and H1975 compared with the normal lung cell line (BEAS-2B). Although miR-1293 expression in LUSC cell line H520 was higher than that in BEAS-2B, we did not find any significant change between H226 and BEAS-2B cells (Fig. 1L). Collectively, these results indicated that miR-1293 was increased in LUAD and elevated miR-1293 might serve as a promising prognostic biomarker for LUAD patients.

Figure 1 miR-1293 expression in human lung cancer tissues and cell lines.

(A) miR-1293 expression in human LUAD tissues analyzed by Starbase online software. (B) miR-1293 expression in human LUSC samples, which was assayed by Starbase online software. (C) The relationship between miR-1293 and the pathologic stage of LUAD patients in TCGA database. The association between miR-1293 and the overall survival of LUAD patients was assayed by linkedomics (D) and starbase (E) software. (F) The relationship between miR-1293 and the pathologic stage of LUSC patients. (G) The association between miR-1293 and the overall survival of LUSC patients. The correlation between miR-1293 expression and the frequent mutations, including KRAS (H), EGFR (I), ALK (J) and PIK3CA (K) present in LUAD patients. (L) miR-1293 expression was measured in LUAD cell lines (A549, H1299, and H1975), LUSC cell line (H226 and H520) and normal lung cell line (BEAS-2B). *P < 0.05 vs the BEAS-2B cells.

miR-1293 promotes the proliferation, migration and invasion of lung adenocarcinoma cells

We further investigated the role of miR-1293 in LUAD cells. The viabilities of H1975 and H1299 cells were significantly promoted after miR-1293 mimic transfection (Figs. 2A and 2B). The numbers of migrated and invasive cells were increased by miR-1293 mimic (Figs. 2C and 2D). Moreover, the mRNA and protein expression levels of MMP2 and MMP9 were elevated by miR-1293 mimic both in H1975 and H1299 cells (Figs. 2E–2G). In contrast, miR-1293 antagomir suppressed the viabilities, decreased cell migration and invasion, and reduced the expression of MMP2 and MMP9 in LUAD cells (Figs. 2A–2G). To exclude the influence of cell proliferation on cell migration and invasion, LUAD cells in the upper chamber were cultured with RPMI 1640 medium without FBS, which were shown in Fig. S1. In general, these data indicated that miR-1293 could facilitate the proliferation, migration and invasion of LUAD cells in vitro.

Figure 2 miR-1293 facilitates the proliferation, migration and invasion of LUAD cells.

CCK-8 assay for the viabilities of H1975 (A) and H1299 (B) cells after transfection with miR-1293 mimic or antagomir (anti-miR-1293) as well as their negative controls. (C) The number of migrating cells was elevated by miR-1293 mimic, but decreased by anti-miR-1293 both in H1975 and H1299 cells, which were analyzed by the transwell assay. (D) The transwell assay was used to measure the invasion of H1975 and H1299 cells. (E)–(G) The protein levels of MMP2 and MMP9 were detected in LUAD cells under different treatments. *P < 0.05 vs the NC mimic group, #P < 0.05 vs the anti-NC group.

PGM5 as a target of miR-1293 in lung adenocarcinoma

To predict the target genes of miR-1293 in LUAD, we analyzed three independent datasets: the TCGA data, which revealed genes negatively associated with miR-1293 expression in LUAD; the potential target genes of miR-1293 analyzed by TargetScan; and differentially expressed genes in LUAD tissues from the GSE118370. Overlap of the three datasets was displayed using a Venn diagram. We found that five genes were common to all three datasets (Fig. 3A). Among these genes, PGM5 was selected, because it has been found to be related to tumor progression in some cancers (Jiao et al., 2019; Sun et al., 2019). As shown in Fig. 3B, PGM5 expression showed an inverse correlation with miR-1293 in LUAD patients. We found that PGM5 expression was decreased in tumor tissues of LUAD patients (Fig. 3C). Lower PGM5 expression was related to the poor overall survival of LUAD patients (Fig. 3D). Furthermore, the mRNA and protein expression levels of PGM5 were reduced in LUAD cell lines compared with BEAS-2B cells (Figs. 3E and 3F).

Figure 3 PGM5 expression was decreased and negatively associated with miR-1293 in LUAD.

(A) Venn diagram of specific genes between predicted miR-1293 target genes from Targetscan, TCGA database which showed genes negatively related to miR-1293 in LUAD and differentially expressed genes from the GSE118370. (B) The relationship between PGM5 and miR-1293 was analyzed in TCGA database by using linkedomics software. (C) PGM5 expression was decreased in LUAD tumor tissues compared with normal samples analyzed by GEPIA2. (D) The association between PGM5 and the overall survival of LUAD patients was determined by GEPIA2. (E) PGM5 mRNA expression was determined in LUAD cell lines and a normal lung cell line (BEAS-2B). (F) Western blot assay for PGM5 protein level in different cell lines. *P < 0.05 vs the BEAS-2B cells.

The binding sites between PGM5 mRNA and miR-1293 are shown in Fig. 4A. Subsequently, we constructed wild-type and mutated PGM5 dual-luciferase reporter vectors to validate whether PGM5 was a target of miR-1293 in LUAD. The data demonstrated that miR-1293 dramatically decreased the luciferase activities of PGM5 by directly bounding to its 3′UTR, without any effect on the luciferase activities of PGM5-MUT vector (Fig. 4B). Additionally, we found that miR-1293 significantly decreased the mRNA and protein levels of PGM5 in H1975 and H1299 cells (Figs. 4C–4E). Taken together, these results suggested that PGM5 was a target of miR-1293 in LUAD.

Figure 4 PGM5 is a target of miR-1293 in LUAD.

(A) Targetscan predicted the binding sites between miR-1293 and PGM5. (B) H1975 cells were co-transfected with PGM5 wild type (PGM5-WT) or mutant (PGM5-MUT) vectors and miR-1293 mimic or control mimic for 48 h; luciferase activities were detected in these cells following different treatments. (C) The mRNA expression of PGM5 was analyzed in H1975 and H1299 cells after transfection with miR-1293 mimic or its control. (D) and (E) PGM protein levels were examined by western blot. *P < 0.05 vs the NC mimic group.

miR-1293 functions via regulating PGM5 expression in lung adenocarcinoma cells

To explore whether miR-1293 inhibited LUAD cells’ proliferation, migration, and invasion via mediating PGM5, we upregulated PGM5 expression in H1299 cells transfected with or without miR-1293 mimic. Western blot assay showed that pcDNA3.1-PGM5 plasmid significantly enhanced PGM5 expression in H1299 cells (Fig. 5A). PGM5 upregulation notably inhibited cell viability and abrogated the promotion of miR-1293 on the viability of H1299 cells (Fig. 5B). Using transwell assays, we found that overexpression of PGM5 decreased the numbers of migratory and invasive cells and reversed the effects of miR-1293 on the migration and invasion of H1299 cells (Figs. 5C and 5D). Also, PGM5 upregulation decreased the expression of MMP2 and MMP9 that was increased by miR-1293 mimic (Figs. 5E and 5F). Therefore, these data further confirmed that miR-1293 functioned via regulating PGM5 expression in LUAD cells.

Figure 5 miR-1293 functions via regulating PGM5 expression in LUAD cells.

(A) H1299 cells transfected with pcDNA3.1-PGM5 plasmid revealed higher PGM5 expression than that with the pcDNA3.1 vector. (B) Cell viability was determined in H1299 cells under different treatments. (C & D) Transwell assays were used to analyze the migration and invasion of H1299 cells. (E & F) The protein levels of MMP2 and MMP9 were measured in H1299 cells. *P < 0.05 vs the pcDNA3.1 group, #P < 0.05 vs the NC mimic group, &P < 0.05 vs the miR-1293 mimic group.

miR-1293 antagomir suppresses LUAD growth and metastasis via negatively modulating PGM5 level in vivo

The role and mechanism of miR-1293 in LUAD were further evaluated in H1975 xenografted mouse model (Fig. 6A). Administration of anti-miR-1293 significantly reduced miR-1293 expression, accompanied by elevated PGM5 expression in tumor tissues (Figs. 6B and 6C). Tumor volume and Ki-67 expression were reduced by anti-miR-1293 (Figs. 6D–6F). Additionally, MMP9 levels were also decreased in mice treated with anti-miR-1293 (Fig. 6F). Taken together, these results indicated that miR-1293 inhibitor could suppress the growth and metastasis of LUAD via negatively modulating PGM5 level in vivo.

Figure 6 miR-1293 antagomir suppresses LUAD growth and metastasis via negatively modulating PGM5 level in vivo.

(A) A timeline for the H1975 xenograft experiment. (B) miR-1293 expression in tumor tissues. (C) PGM5 protein level was detected in mouse tumor samples by western blot. (D) Mice were sacrificed at 5 weeks after inoculation of H1975 cells, and tumor images were photographed. (E) Tumor volumes were measured every week after inoculation. (F) Immunohistochemical staining for Ki-67 (upper) and MMP9 (lower) in mouse tumor sections.

Discussion

Accumulating evidence has demonstrated that a variety of miRNAs are associated with the progression of lung cancer (Liu et al., 2020a; Song et al., 2020). miR-1293 expression was dramatically upregulated in gingivo buccal cancer samples as well as two types of precancer (leukoplakia and lichen planus) tissues (De Sarkar et al., 2014). It was also increased in pancreatic tumor tissues when compared to adjacent non-tumor pancreatic tissues. High expression of miR-1293 showed a reduced overall survival of pancreatic adenocarcinoma patients (Shi et al., 2018). In renal cell carcinoma, miR-1293 was highly expressed, and patients with higher miR-1293 had an unfavorable prognosis. Upregulation of miR-1293 promoted cell viability, migration and invasion in renal cell carcinoma cells (Liu et al., 2020b). However, Takagawa et al. (2020) found that miR-1293 was a tumor-suppressive miRNA and it suppressed colon cancer cell growth both in vitro and in vivo. Currently, its role in lung cancer, especially in LUAD remains poorly understood.

Recent studies have revealed that miR-1293 is an independent prognostic factor in LUAD patients and it is involved in the survival of these patients (Wang et al., 2020; Zheng et al., 2018). Consistent with these results, we found miR-1293 expression was associated with the pathologic stage of LUAD patients, and patients with higher miR-1293 expression showed poor overall survival. Analyzing the expression level of miR-1293 in LUAD patients in TCGA database indicated that miR-1293 expression was upregulated in LUAD samples when compared with normal samples. Interestingly, we found that miR-1293 expression was positively related to KRAS and PIK3CA mutations, but negatively related to EGFR mutation in LUAD patients. Similarly, there was an increase of miR-1293 in LUAD cell lines relative to a normal lung cell line. Hence, we hypothesized that miR-1293 might be a tumor promotor in LUAD. In this study, we also found miR-1293 expression was elevated in human LUSC samples. However, its level was not associated with patients’ pathologic stage and overall survival. Although miR-1293 expression in LUSC cell line H520 was higher than that in BEAS-2B, we did not find significant change between H226 and BEAS-2B cells, this difference may be due to cell lines were established in different ways. Given that miR-1293 expression was not related to the clinical characteristic of LUSC patients, we did not further investigate the role and mechanism of miR-1293 in LUSC.

Subsequently, we investigated the effects of miR-1293 on the malignant phenotypes of LUAD cells. In vitro experiments verified that miR-1293 could promote the proliferation, migration and invasion of LUAD cells. On the contrary, miR-1293 antagomir suppressed these characteristics of LUAD cells. Moreover, we found that miR-1293 antagomir reduced tumor growth and Ki-67 staining in a xenograft mouse model. MMP2 and MMP9 are important proteolytic enzymes that degrade collagen type IV, the main component of the basement membrane. They are often hyper-expressed during cancer progression, and related to tumor invasion and metastasis (Shang et al., 2019; Yang et al., 2017; Zhao et al., 2015). In this study, we found that miR-1293 mimic significantly upregulated MMP2 and MMP9 expression, but their levels were reduced by miR-1293 antagomir in LUAD cells. Likewise, MMP9 expression was downregulated by miR-1293 antagomir in tumor tissues of xenografted mice. This further confirmed that miR-1293 could promote the migratory and invasive capabilities of LUAD cells. Taken together, these data demonstrated that miR-1293 was a tumor promotor that facilitated the proliferation, migration and invasion of LUAD cells.

The mechanism of miR-1293 in LUAD was also investigated. Phosphoglucomutases (PGMs) are a family of metabolic enzymes that catalyze the bidirectional interconversion of glucose-1-phosphate and glucose-6-phosphate. It has been demonstrated that PGMs are involved in the proliferation, invasion and metastasis of cancer (Jin et al., 2018; Lee et al., 2010; Ricciardiello et al., 2018). PGM5, which is also known as phosphoglycosidase-related protein (PGM-rp) and Aciculin, is located on human chromosome 9 (9q21.11) (Edwards et al., 1995). In 1995, Edwards et al. (1995) identified that PGM5 is a member of the PGM family. Recent studies found that PGM5 expression is decreased in liver cancer and colorectal cancer, and low PGM5 levels are associated with patients’ poor overall survival (Jiao et al., 2019; Sun et al., 2019). As well, PGM5 is reported to inhibit the proliferation, invasion and migration abilities of colorectal cancer cells (Sun et al., 2019). By analyzing TCGA database, we found PGM5 expression was decreased and negatively involved in miR-1293 in LUAD patients. Lower PGM5 expression was related to the poor overall survival of LUAD patients. Importantly, Targetscan software predicted that PGM5 was a potential target of miR-1293. In this study, we found that PGM5 overexpression could inhibit the proliferation, migration, invasion and expression of MMP2 and MMP9 in LUAD cells. This indicated that PGM5 was a tumor suppressor in LUAD. Using a dual-luciferase assay, we verified that PGM5 was a target of miR-1293 in LUAD cells. miR-1293 could suppress the expression of PGM5 in H1299 and H1975 cells. To further investigate whether miR-1293 functioned via regulating PGM5, we upregulated PGM5 in miR-1293 mimic-transfected H1299 cells and found that PGM5 upregulation reversed the effects of miR-1293 mimic on the proliferation, migration, invasion and expression of MMP2 and MMP9 in this cell line. Moreover, this study found that anti-miR-1293 could elevate PGM5 expression in H1975 xenograft mouse model. In general, these data suggested that miR-1293 functioned via targeting PGM5 in LUAD.

Conclusions

Our study found that elevated miR-1293 could promote the proliferation, migration and invasion of LUAD cells. Mechanism exploration verified that miR-1293 played these effects in LUAD via targeting PGM5. This study illustrates comprehensive insights into the roles of miR-1293 and PGM5 in LUAD, which may serve as potential therapeutic targets for LUAD patients.

Supplemental Information

Supplemental Information 1 ARRIVE 2.0 checklist.

Click here for additional data file.

Supplemental Information 2 Uncropped Blots.

Click here for additional data file.

Supplemental Information 3 Raw numerical data.

Click here for additional data file.

Supplemental Information 4 Cell proliferation in serum-free medium.

H1975 and H1299 cells transfected with miR-1293 mimic or antagomir (anti-miR-1293) were seeded into 96-well plates and cultured in serum-free RPMI 1640 medium for 48 h, cell viability was analyzed by CCK-8 assay.

Click here for additional data file.

Additional Information and Declarations

Competing Interests

Author Contributions

Animal Ethics

Data Availability

The authors declare that they have no competing interests.

Bing Chen conceived and designed the experiments, performed the experiments, analyzed the data, prepared figures and/or tables, authored or reviewed drafts of the paper, and approved the final draft.

Shiya Zheng analyzed the data, authored or reviewed drafts of the paper, and approved the final draft.

Feng Jiang conceived and designed the experiments, authored or reviewed drafts of the paper, and approved the final draft.

The following information was supplied relating to ethical approvals (i.e., approving body and any reference numbers):

All procedures involving in animals were reviewed and approved by the Affiliated Cancer Hospital of Nanjing Medical University (No. 2020-005).

The following information was supplied regarding data availability:

The raw data are available in the Supplemental Files.

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
