# Peer review of "miR-1293 acts as a tumor promotor in lung adenocarcinoma via targeting phosphoglucomutase 5"

_PeerJ, doi:10.7717/peerj.12140_

## Round 0.1 · original submission · Minor Revisions

Please address all the reviewers' comments as by doing so the presentation of the manuscript will be greatly enhanced. Additionally to those comments please address the following:

1) Does miR-1239 expression correlate with the frequent mutations present in lung adenocarcinoma patients (eg. KRAS, EGFR, ALK... etc) ? In other words is their a particular subtype of lung adenocarcinoma for which miR-1239 is more clinically relevant? Such data would enhance figure 1 and the discussion.

2) The migration and invasion assay results have a confounding factor: proliferation. Due to the fact that such experiments were evaluated 48 hours after cell seeding, and according to data in Figures 2A and B that gives enough time for the cells to proliferate. The experiments should be redone at shorter time intervals or with the use of proliferation inhibitors such as mitomycin.

Reviewer 1 ·

Basic reporting

In the manuscript, Chen et al elucidate that miR-1293 promotes proliferation, migration and invasion of LUAD cells via targeting PGM5, thereby suggesting its implication as potential therapeutic target for LUAD patients. They also showed that LUAD patients have elevated miR-1293 and decreased PGM5 expression with the miR-1293 expression negatively affecting patient's survival.


Overall, this is a strong comprehensive study that elucidates that miR-1293 promotes tumor in LUAD. The language is clear and easy to follow. Proper background/context has been provided where relevant. The conclusions are well supported by experimental evidences. However, I feel the quality of some of the figures need to be improved, before the manuscript is ready for publication. I have some of my other recommendations below.

Experimental design

The experiments are well executed with proper controls. The questions are logical and the methods described in sufficient details.

I have the following suggestion:
1. Some of the figures (1A-G) need to be relabeled with bigger fonts, i think the figures are screenshots of graphs prepare by the software used, all legends in the graph need to be written in bigger legible font.
2. Do the authors have an explanation for why they were not able to see a high miR-1293 expression in LUSC cells compared to normal (1H) although they were reported to be higher on patients (1B) ? Did the authors check other LUSC cell lines (recommended)?
3. For some of the blots, a lower exposure would be better to understand the difference in protein levels (example- 3F,4D, 5A PGM5)
4. For figure 6, a timeline for the xenograft experiment will be helpful.

Validity of the findings

I found all the conclusions of the authors sound and logical.
I am curious if the author's conclusion that miR-1293 acts as a tumor promotor by targeting phosphoglucomutase 5 is unique to LUAD (What about LUSC?) and hope the authors find a way to address this either in the experiment or in the discussion section. This is especially an issue since they showed its expression is high in both LUAD and LUSC patients (1A and B), but dont show the PGM5 expression in LUSC in the later experiments.

Additional comments

There are some grammatical errors in the manuscript. Please have them proofread before final submission.

·

Basic reporting

In their report, Chen et. al. analyses the expression of hsa-miR-1293 in lung adenocarcinoma tumors and explore its potential functional role on proliferation, migration and invasion of cancer cell. They finally provides In vitro and In vivo evidence that hsa-miR-1293 targets PGM5 gene, which is proposed as a plausible molecular mechanism trough which exerts its anti-neoplastic roles

Overall, the manuscript conforms to Journal's standards. It is well written, with a clear syntax and professional style. Justification for the study is properly established in the background and intro. Some minor details I still detect, includes few misspellings and syntax errors along the text, which I would invite the authors to revise. Some of them can be located in lines 27, 53, 54, 248.

In general, data provided in the form of figures and supplementary data are useful and makes it easier for the reader to follow the results and findings. An exception to this is Figure 1. I understand some of these plots have been automatically generated by the online bioinformatics resources reviewed by the authors. Still, quality of images are of low resolution and plot titles and legends are very difficult to read. This aspect might be improved. Perhaps, authors might consider distributing plots into two separated figures.

Experimental design

Methodology is properly described, in line with the research question and objectives.

For the greatest part, procedures are described with sufficient detail an providing information for further replication. However, a minor number of procedures/findings require further description or improved clarity in order to fully permit other researchers replicate authors findings

Regarding the Bioinformatics analysis section. Here, authors claim the have employed three resources (i.e., Starbase, GEPIA2 and Linkedomics), nonetheless, in the paragraph beginning at line 182 of their results, they mention that "TargetScan" and a dataset "GSE118370", were also explored. I consider, details about what these resources are (e.g., Version, study design of dataset, etc), how they were retrieved should be briefly described in methods section.

On this same aspect, results description and discussion in line 273 could be improved, given that in its current state remains somewhat ambiguous. With particular regard to the sentence "Venn diagram was used to predict the target genes of miR-1293 from three independent database" in line 187, which I fear does not exactly convey what the authors intended, which I believe is the fact that they looked at the intersection of the results from 3 different exploration strategies to find putative miRNA targets (i,e., bioinformatic prediction of seed regions, evidence of anti correlation in expression, etc). In this context, venn diagram simply serves the purpose of visual representation.

Another point of discussion regards the functional studies. H1299 cell line was used to study the regulation exerted by miR-1293 over PGM5 expression and H1975 cell line was used instead to perform functional studies in vivo. A brief discussion of the rationale behind this decisions would add value the research.

Validity of the findings

I consider that the characterization of of miR-1293 expression alteration and its roles at suppressing proliferation, migration and invasion of LUAD cells and its anti tumorigenic potential are well established and supported by the underlying data in this study. The discussion unfolds concisely and conclusions are congruous with original research question and restricted to the observations and supporting results.

Additional comments

I commend the authors for their work which is well presented in a concise and well supported way

---

## Round 0.2 · accepted · Accept

All issues pointed by the reviewers were adequately addressed and the manuscript was revised accordingly. Therefore it is acceptable now.

Reviewer 1 ·

Basic reporting

No commeny

Experimental design

No comment

Validity of the findings

No comment

Additional comments

The authors have addressed all my concerns and the manuscript is ready to be published now.

·

Basic reporting

No comment

Experimental design

The authors have properly addressed my previous comments and questions.

In light of the new data presented by the authors, I observe a pair of minor details regarding the results which could be improved. I enlist them below.

Regarding the correlations between miR-1293 and somatic mutations. Is the shown p-value related to a correlation test (e.g., Pearson, Spearman) or is it the result of a t-test. Clarifying this information in the Methods and/or Results sections would be very helpful.

Similarly, survival analysis curves produce with bioinformatic public interfaces was most likely done using a Log rank Mantel cox test or similar test. Please clarify this in Methods or Results sections

Validity of the findings

no further comment